# Isolation and Identification of a *TaTDR-Like* Wheat Gene Encoding a bHLH Domain Protein, Which Negatively Regulates Chlorophyll Biosynthesis in *Arabidopsis*

**DOI:** 10.3390/ijms21020629

**Published:** 2020-01-17

**Authors:** Yu Xia, Zheng Li, Junwei Wang, Yanhong Li, Yang Ren, Jingjing Du, Qilu Song, Shoucai Ma, Yulong Song, Huiyan Zhao, Zhiquan Yang, Gaisheng Zhang, Na Niu

**Affiliations:** 1Key Laboratory of Crop Heterosis of Shaanxi Province, Wheat Breeding Engineering Research Center, Ministry of Education, College of Agronomy, Northwest A & F University, Yangling 712100, Shaanxi, China; xiayu325@nwafu.edu.cn (Y.X.); lizheng9045@nwafu.edu.cn (Z.L.); wjw@nwsuaf.edu.cn (J.W.); yanhong1417@163.com (Y.L.); yangren1028@nwafu.edu.cn (Y.R.); dou0321@nwafu.edu.cn (J.D.); qilusong@nwafu.edu.cn (Q.S.); mashoucai@sohu.com (S.M.); sylbl1986@nwsuaf.edu.cn (Y.S.); zhqyang6688@nwsuaf.edu.cn (Z.Y.); 2College of Plant Protection, Northwest A & F University, Yangling 712100, Shaanxi, China; zhaohy@nwsuaf.edu.cn

**Keywords:** bHLH transcription factor, chlorophyll biosynthesis, myc family, *TaTDRL*, wheat

## Abstract

Chlorophyll biosynthesis plays a vital role in chloroplast development and photosynthesis in plants. In this study, we identified an orthologue of the rice gene *TDR* (*Oryza sativa* L., Tapetum Degeneration Retardation) in wheat (*Triticum aestivum* L.) called *TaTDR-Like* (*TaTDRL*) by sequence comparison. TaTDRL encodes a putative 557 amino acid protein with a basic helix-loop-helix (bHLH) conserved domain at the C-terminal (295–344 aa). The TaTDRL protein localised to the nucleus and displayed transcriptional activation activity in a yeast hybrid system. *TaTDRL* was expressed in the leaf tissue and expression was induced by dark treatment. Here, we revealed the potential function of *TaTDRL* gene in wheat by utilizing transgenic *Arabidopsis* plants TaTDRL overexpressing (TaTDRL-OE) and TaTDRL-EAR (EAR-motif, a repression domain of only 12 amino acids). Compared with wild-type plants (WT), both TaTDRL-OE and TaTDRL-EAR were characterized by a deficiency of chlorophyll. Moreover, the expression level of the chlorophyll-related gene *AtPORC* (*NADPH:protochlorophyllide oxidoreductase C*) in TaTDRL-OE and TaTDRL-EAR was lower than that of WT. We found that TaTDRL physically interacts with wheat Phytochrome Interacting Factor 1 (PIF1) and Arabadopsis PIF1, suggesting that TaTDRL regulates light signaling during dark or light treatment. In summary, TaTDRL may respond to dark or light treatment and negatively regulate chlorophyll biosynthesis by interacting with AtPIF1 in transgenic *Arabidopsis*.

## 1. Introduction

Leaf photosynthesis is the basis for the formation of grain in the production of food crops. Chlorophyll is an important biomolecule for the maintenance of plant life activities. The accumulation and degradation of chlorophyll are also associated with chloroplast development, photomorphogenesis, and chloroplast-nuclear signaling [1]. Chlorophyll synthesis has three main steps: chlorophyll biosynthesis, chlorophyll cycle (interconversion of chlorophyll a and chlorophyll b), and chlorophyll degradation [1,2].

Five photoreceptors have been identified in *Arabidopsis*, including the phytochromes, cryptochromes, phototropins, UV Resistance Locus 8 (UVR8), and the ZTL/FKF1/LKP2 family of F-box proteins [3,4,5]. Among them, the phytochrome family of photoreceptors, which maximally absorbs blue, red and far-red lights, are encoded by five different genes (PHYA to PHTE) in *Arabidopsis* [3,6]. Phytochromes are also responsible for regulating seed germination, seedling growth, shade avoidance, flowering, and other adaptive responses [3]. In the nucleus, phytochromes interact with a small subset of basic helix-loop-helix (bHLH) transcription factors called Phytochrome Interacting Factors (PIFs) [7,8]. These PIFs have been shown to repress seed germination, chlorophyll accumulation and the assembly of photosynthetic complexes, as well as promotion of seedling skotomorphogenesis and shade-avoidance [8,9]. Among these PIFs, PIF1 functions as a negative regulator of chlorophyll biosynthesis in the dark and regulates gibberellic acid biosynthesis and sensitivity to control seed germination [10]. *pif1* mutant seedlings accumulate higher amounts of free protochlorophyllide (Pchlide), a phototoxic intermediate in the chlorophyll biosynthetic pathway, in the dark [11]. Subsequent light exposure causes photooxidative damage and bleaching of pif1 seedlings [3,8]. PIF1 shows transcriptional activation activity in the dark, which is reduced by light-induced degradation of PIF1 to promote chlorophyll biosynthesis and seed germination in light [3].

PIFs belong to the bHLH superfamily of transcription factors [12]. The bHLH domain is a DNA-binding and dimerization domain that allows the formation of homo- and/or heterodimers. The bHLH proteins are transcriptional regulators that function either as activators or as repressors. For example, PIF3 can homodimerize and heterodimerize with PIF4 [13]. Furthermore, PIF3 also can interact with a non-PIF bHLH factor Long Hypocotyl in Far-Red 1 (HFR1) [14]. In addition, the dimer of bHLH transcriptional factors can bind to *cis*-acting elements found in the promoter regions of target genes. Each member of the dimer binds the DNA region through its basic domain. The most common of these cis-elements are G-box (CACGTG) and E-box (CANNTG). PIFs preferentially bind a G-box DNA sequence element, which is a subclass of an E-box element present in many light-regulated promoters [1,2]. For example, the PIF3-PIF3 homodimer and the PIF3-PIF4 heterodimer can bind to a G-box DNA sequence element in vitro [13,14,15]. PIF1 has been shown to directly bind to the promoter of the *PORC* (NADPH:protochlorophyllide oxidoreductase C) gene and also to indirectly regulate other chlorophyll biosynthetic genes [10]. Moreover, PIF1 activates expression of *PORC* in a G-box dependent manner. These data suggest that PIF1 directly or indirectly regulates key genes involved in chlorophyll biosynthesis to optimize the greening process in *Arabidopsis*.

MYC proteins are a family of regulatory genes that encode essential nuclear transcription factors and belong to the superfamily of bHLH DNA-binding proteins. The bHLH proteins consist of a superfamily of transcription factors (TFs) found in both plants and animals. A total of 147 and 167 bHLH-TFs were predicted in the *Arabidopsis* and Oryza sativa genome, respectively [13,16]. In previous studies, the *Oryza sativa* L. Tapetum Degeneration Retardation (*OsTDR)* gene is preferentially expressed in the tapetum and encodes a putative basic helix-loop-helix protein, which is localized to the nucleus. *OsTDR* gene is involved in a crucial regulation network controlling postmeiotic anther development [17]. However, no related research on *TDR* genes involved in chlorophyll biosynthesis has been reported. In the study, we identified an orthologue of the rice gene *OsTDR* in wheat called *Triticum aestivum* L. Tapetum Degeneration Retardation-Like *(TaTDRL)*. We investigated the expression pattern of *TaTDRL* and its response to dark or light treatment as well as its role in chlorophyll biosynthesis by overexpressing *TaTDRL* in transgenic plants. To further investigate the function of TaTDRL, we generated TaTDRL-EAR transgenic plants by using chimeric repressor silencing technology (CRES-T) [18]. Phenotypic analyses indicated that TaTDRL decreased the content of chlorophyll in both of two transgenic plants, which exhibited decreased chlorophyll biosynthesis. Furthermore, TaTDRL physically interacts with AtPIF1 and TaPIF1 in yeast two-hybrid assays. Taken together, these results suggested that TaTDRL may participate in the negative regulation of chlorophyll biosynthesis by interacting with TaPIF1 in wheat.

## 2. Results

### 2.1. Identification and Sequence Analysis of TaTDRL

In the present study, we cloned *TaTDRL* (TraesCS6D02G069300.1) gene from wheat cultivar Xi Nong 1376 (XN1376) and the full-length cDNA comprising 2150 bp. Its complete coding sequence (CDS) is 1674 bp in length and encodes a protein of 557 amino acid redidues, with a predicted molecular mass of 59.21 kDa and a pI of 4.63. The 5′ and 3′ untranslated regions (UTRs) were 251 bp and 225 bp (Figure 1A), respectively. The genomic organization of *TaTDRL* is shown in Figure 1B and there are seven exons and six introns. The bHLH conserved domain spans amino acids 295–344 (Figure 2). Alignment of the complete protein sequence with those of plant homologs demonstrated that TaTDRL is identical to transcription factor TDR-Like of *Aegilops tauschii* (100%). Based on this alignment, a phylogenetic tree was constructed (Figure 3). It is similar to OsTDR, and homologs from *Hordeum vulgare* (BAK06269.1), *Setaria italica* (XP.012700935.1), and *Arabidopsis thaliana* aborted microspores (AMS). The grand average of hydropathy (GRAVY) value of TaTDRL was −0.495, which indicates that TaTDRL is hydrophilic protein (Appendix A). The secondary structure prediction of TaTDRL protein showed that the protein contained 36.62% α-helices, 50.09% random coils, 9.16% extended strands, and 4.13% β-turns (Appendix A).

### 2.2. Expression Patterns and Promoter Analysis of TaTDRL

The expression patterns of *TaTDRL* in different organs were analyzed using quantitative real time polymerase chain reaction (qRT-PCR). Results showed that the expression levels of *TaTDRL* were high in the leaf and panicle, with the highest expression levels in the latter. By contrast, *TaTDRL* was expressed at very low levels in the grain, root, stem (Figure 4A). We analyzed the regulatory elements in the *TaTDRL* promoter region using the PlantCARE and NewPLACE databases. Several regulatory DNA motifs were found in the *TaTDRL* promoter, including cis-acting elements involved in phytohormone signaling, abiotic stress responses, and light responsiveness (Appendix A). In particular, 16 cis-elements associated with light responsiveness were identified (Figure 4B). Under light treatment, the transcript levels of *TaTDRL* decreased at 3 h, reach its minimum at 6 h ). This indicated that light treatment considerably reduced *TaTDRL* expression. Under dark treatment, the expression of *TaTDRL* increased at 3 h, reach its maximum at 6 h, and returned to normal levels at 12 h and 24 h, respectively (Figure 4C). These results further illustrated that *TaTDRL* transcript levels might be regulated by light or dark signals.

### 2.3. Subcellular Localization and Transactivation Activity of TaTDRL

An Agrobacterium harboring a recombinant construct expressing TaTDRL-enhance green fluorescence protein (EGFP) fusion protein or EGFP alone was transfected into tobacco epidermal cells to confirm the subcellular localization of TaTDRL. Nuclear localization was observed in TaTDRL-EGFP transformed cells, while the signal of control free EGFP was distributed in the whole cell, (Figure 5A). To investigate if TaTDRL has a transcriptional activation activity, the CDS of *TaTDRL* was fused to the GAL4 DNA-binding domain (GAL4-BD) and the resulting construct was transformed into yeast strain Y2H. The reporter genes of *HIS3*, *ADE2*, and *MEL1* were expressed in the yeast cells transformed with the TaTDRL-BD construct, suggesting that TaTDRL has a transcriptional activation activity. A series of truncations in both the N- and C termini of TaTDRL were produced. Assays of these truncated proteins revealed that the N-terminal region (1–294 aa) was required for transactivation activity of the fusion protein in yeast (Figure 5B). These results indicated that TaTDRL functions in the nucleus as a transcription factor.

### 2.4. Effect of TaTDRL Overexpression in Arabidopsis Decreased the Content of Chlorophyll

TaTDRL was overexpressed in *Arabidopsis* to confirm the function of this gene in chlorophyll biosynthesis. Two independent homozygous T3 transgenic lines were used for analysis (OE1 and OE2). Expression levels of *TaTDRL* were confirmed using qRT-PCR; *TaTDRL* was highly expressed in the two positive lines, but not expressed in the WT (Figure 6A). After seven days, differences in plant size were observed. Under normal growth conditions, there were larger leaves in the overexpressing lines (Appendix A). Compared with WT plants, leaf whitening was seen in almost all leaves of TaTDRL transgenic plants at two weeks post-germination. For three-week-old plants, the color of transgenic *Arabidopsis* leaves was still lighter than WT (Appendix A). On the 28th day, no major difference in plant growth were identified, except for early flowering time in the overexpressing lines (Figure 6B). We further evaluated chlorophyll content in leaves (Figure 6C). Both WT and TaTDRL-OE plants accumulated maximum chlorophyll in the third week of growth. However, compared with WT plants, the two transgenic lines showed lower chlorophyll content at each time point.

### 2.5. Transgenic Plants Expressing the Chimeric TaTDRL Repressor

TaTDRL-EAR transgenic lines were constructed. As that the EAR-motif (LDLDLELRLGFA) has been shown to convert a transcriptional activator into a strong repressor, and that the repressive activity of the EAR-motif repression domain was dominant over both intra- and intermolecular activational activities [18]. We used an alternative approach to fuse the coding region of TaTDRL with the DNA encoding the EAR-motif repression domain (RD) (Appendix A). Then the intact plant transcriptional repression domain was converted to suppress the expression of the target genes of TaTDRL in stably transformed *Arabidopsis*. According to the previous description, two independent homozygous T3 transgenic lines were used for analysis (EAR-1 and EAR-2) (Figure 6A). Both of the transgenic plants and the WT were grown for four weeks under normal conditions. Compared with WT, the transgenic lines showed inhibition of growth in rosettes (Appendix A). On the seventh day, there were obvious differences in plant size and the number of leaves. It is remarkable that most of transgenic lines have smaller and fewer leaves than WT (Appendix A). The color of the leaves from the second week to the third week is lighter than WT (Appendix A). On the 28th day, no differences were observed, except that both transgenic lines were smaller (Figure 6B). Under normal conditions, the chlorophyll content of the TaTDRL-EAR plants was much lower than that of the WT control between one to four weeks growth (Figure 6C; Appendix A).

### 2.6. TaTDRL-OE and TaTDRL-EAR Affect the Expression of a Chlorophyll-Related Gene

AtPORC encodes a protochlorophyllide oxidoreductase, which catalyzes a key step in chlorophyll biosynthesis and directly affects chlorophyll accumulation. The transcription levels of *AtPORC* was compared in WT, TaTDRL-OE and TaTDRL-EAR plants by qRT-PCR to elucidate the molecular mechanism related to TaTDRL in chlorophyll biosynthesis (Figure 6D). Under normal conditions, the transcription level of *AtPORC* was significantly lower in the TaTDRL-OE and TaTDRL-EAR lines than the controls at every time point. This results agreed with the phenotype results obtained from the TaTDRL-OE and TaTDRL-EAR plants grown in soil. Furthermore, to ascertain whether TaTDRL can directly interact with the proteins encoded by these genes (AtPIF1, TaPIF1), protein-protein interactions were identified using the yeast two-hybrid system. AtPIF1 and its homologous protein TaPIF1 both interacted with TaTDRL (Figure 7A,B). In addition, in-vivo Bimolecular Fuorescence Complementation (BiFC) assays in tobacco epidermal cell confirmed that the interactions between TaTDRL and AtPIF1 and between TaTDRL and AtPIF1 take place in the nucleus (Figure 7C). The interaction of TaTDRL and AtPIF1 provides additional evidence for the involvement of TaTDRL in chlorophyll biosynthesis. TaTDRL may be a direct negative regulator of the chlorophyll biosynthesis in transgenic *Arabidopsis*.

## 3. Discussion

### 3.1. Structure and Localization of the TaTDR-Like Gene

Here we isolated a MYC type transcription factor TaTDRL in wheat. TaTDRL encodes a bHLH conserved domain at the C-terminal (295–344 aa), which localized to the nucleus and displayed transcriptional activation activity. MYC class transcription factors regulate genes involved in growth, cell cycle, signaling, and adhesion [19,20]. In generally, the carboxyl terminus of MYC family members contains a bHLH leucine zipper motif (bHLH-Zip), which has DNA-binding activity and is proposed to form homodimers or heterodimers with MYC-associated factor X (MAX) proteins via their bHLH domain [21,22]. In this study, we found that TaTDRL physically interacts with two other MYC transcription factors, AtPIF1 and TaPIF1, through its C-terminal region without a bHLH domain, and the bHLH region of TaTDRL cannot interact with them. These results indicate that the C-terminus of TaTDRL plays an important role in protein interaction, which is similar to previous research [23].

The MYC/MAX heterodimers bind variants of the E-box motif “CANNTG” through bHLH domain, which can be found in promoters or transcribed sequences of MYC target genes and such binding usually activates the target gene [24,25]. The bHLH domain is involved in anthocyanin biosynthesis, phytochrome signaling, fruit dehiscence, and carpel and epidermal development, as well as environmental stress responses [13,26,27,28,29,30,31,32]. In our study, we found that TaTDRL can form heterodimers with AtPIF1 or TaPIF1, suggesting that in both *Arabidopsis* and wheat, these heterodimers may be involved in important developmental events, such as phytochrome signaling and chlorophyll biosynthesis. We found that the N-terminal region (1–294 aa) is necessary for transactivation activity of TaTDRL in yeast (Figure 5B) through a series of assays of truncated proteins. Each region of TaTDRL has different functions, for example, the N-terminus is responsible for transactivation regulation, bHLH is responsible for DNA-binding or dimerization, and the C-terminal region is responsible for protein interaction. These results provide a theoretical basis for further study of the function of TaTDRL or other MYC transcription factors.

### 3.2. TaTDRL May Be a Direct Negative Regulator of Chlorophyll Biosynthesis

Chlorophyll synthesis or degradation in plants and transformation of proplastids into mature chloroplasts are regulated by both nuclear and chloroplast genes. Mutations in these genes result in visible leaf color variations, mainly due to chlorophyll biosynthesis or chloroplast developmental abnormalities. TaTDRL is homologous to OsTDR and AtAMS in evolutionary relationships, and shares a highly conserved bHLH domain with them and, TaTDRL and OsTDR have more similar structure in regions other than bHLH domain than either does to AtAMS. This suggests that TaTDRL is functionally similar to OsTDR. In rice, previous studies have shown that loss of function of OsTDR affects the expression of the genes encoding chloroplast components, but the specific mechanism has not been well studied [33]. In our study, we found that TaTDRL can inhibit chlorophyll biosynthesis in TaTDRL-OE plants, but we do not know whether this inhibition is direct or indirect inhibition. The CRES-T, exploiting the EAR-motif repression domain is a simple and effective tool to overcome genetic redundancy, which is widely used in research related to plant development [18]. For activating transcription factors, it can phenocopy their corresponding loss-of-function mutants, while for repressing transcription factors, it can investigate their inhibitory effect [34,35]. In order to investigate whether TaTDRL can act as a direct transcriptional repressor, we fused the EAR repressor domain to the TaTDRL coding sequence. The resulting TaTDRL-EAR plants have decreased chlorophyll content. The similarities between the phenotypes of the TaTDRL-OE and TaTDRL-EAR lines suggest that TaTDRL can act as a direct inhibitor of chlorophyll synthesis. The decreased expression of *AtPORC*, which catalyzes a key step in chlorophyll biosynthesis directly affects chlorophyll accumulation [36]. In our study, we found that the expression of *AtPORC* was decreased in both TaTDRL-OE and TaTDRL-EAR lines, which supports the fact that chlorophyll content is reduced in the two transgenic plants. Previous research has shown that PIF1 negatively regulates chlorophyll biosynthesis and seed germination in the dark, and light induced degradation of PIF1 relieves this negative regulation to promote photomorphogenesis [10]. In *Arabidopsis*, PIF1 regulates expression of *PORC*, while PIF1 directly binds to a G-box (CANNTG) DNA sequence element present in the *PORC* promoter, involved in controlling the chlorophyll biosynthetic pathway [10]. In our research, we have also confirmed that TaTDRL can physically interact with AtPIF1 to form a heterodimer and inhibit chlorophyll biosynthesis by inhibiting the expression of *PORC* in transgenic *Arabidopsis*. In addition, we found that there are many light responsive *cis*-elements in the promoter of TaTDRL, and the expression of TaTDRL in wheat seedlings is also affected by dark or light treatment. Our research also confirmed that TaTDRL can also physically interact with TaPIF1. Therefore, based on these results and the above results from transgenic *Arabidopsis*, we hypothesized that TaTDRL is able to respond to light signals and then fine-tune chlorophyll synthesis in wheat, the mechanism of which is similar to that in *Arabidopsis*. Further, an in-depth functional analysis of *TaTDRL* gene using transgenic wheat will be necessary to prove its complete function.

## 4. Materials and Methods

### 4.1. Plant Materials and Growth Conditions

Wheat cultivar XN1376 was used in this study. Seeds were sown in chambers and managed under the normal condition (24/17 °C, 60% humidity and 14/10 h (light/dark)). Seedlings of XN1376 watered daily to avoid drought stress. Seedlings were grown to the two-leaf stage and then subjected dark or light treatment. For determination of the expression pattern of *TaTDRL* under dark and light treatments, the XN1376 seedlings were treated with dark and light for 0–24 h, respectively. The leaves were harvested at 0, 3, 6, 9, 12, and 24 h during treatment. For the tissue specific expression analysis, XN1376 seeds were sown in the experimental field (108°82′ E, 34°15′ N) to obtain grains, roots, stems, leaves and panicles. All samples were immediately frozen in liquid nitrogen and stored at −80 °C for further analysis. All experiments were repeated three times.

*Arabidopsis* thaliana ecotype Columbia (Col-0) was derived from our laboratory. The seeds were surface sterilized in a solution 10% NaClO for 8 min and washed with sterilized water for five times. Seeds were sown on 1/2MS medium [Murashige and Skoog, 0.9% (*w*/*v*) agar, 1% (*w*/*v*) sucrose, pH 5.9]. After three days of stratification at 4 °C, the plates were transferred to a plant growth incubator for seven days before transfer to soil under a 16 h light/8 h dark photoperiod at 22 °C in a growth room.

### 4.2. Cloning and Sequence Analysis of TaTDRL Gene

The OsTDR protein sequence was obtaind from GenBank accession (XP_015625730.1). Database searches of the nucleotide and deduced amino acid sequences of *TaTDRL* were performed by reference genomic database at NCBI/EnsemblPlants/Blast. The predicted gene with highest score and lowest E-val was selected as targeted gene. The CDS sequence of targeted gene was acquired from EnsemblPlants database, and was confirmed in NCBI database. The full-length CDS of *TaTDRL* was amplified using the following gene-specific primers: TaTDRL-P1-F: 5′-ATGGGAGGAGGAGATTATCACC-3′ and reserve primer TaTDRL-P1-R: 5′-TCAATCCATGGCGAGGTACTGC-3′. Total RNA was extracted from leaves of XN1376 plants using RNAiso Plus (TaKaRa, Kyoto, Japan). First-strand cDNA was synthesized using a PrimeScript RT Reagent Kit with gDNA Eraser (TaKaRa, Japan). PCR amplification was carried out for 3 min at 94 °C, followed by 44 cycles of 98 °C for 10 s, 64 °C for 30 s, and 68 °C for 90 s with a final extension at 68 °C for 7 min. The PCR fragment was inserted into the pClone007 sequencing vector and confirmed by sequencing (TSINGKE, Beijing, China). To analyze the gene structure of *TaTDRL*, KOD-Plus-neo (TOYOBO, Osaka, Japan) was used to amplify the genome sequences (The primers used are listed in Appendix A).

### 4.3. Bioinformatics Analysis of TaTDRL

The protein sequence of TaTDRL (TraesCS6D02G069300.1) is available at the EnsemblPlants website (http://plants.ensembl.org/index.html). The online tool SMART (http://smart.embl-heidelberg.de/) was used for conserved domains analysis of TaTDRL. Alignments were obtained with ClustalX and TEXshade. Logo diagrams used to define consensus sequences were obtained using multiple sequence alignments for TaTDRL, OsTDR, and AtAMS by TEXshade [37]. A phylogenetic tree was constructed using ClustalX and MEGA X. Prediction of GRAVY was performed using the online tool ProtScale (http://www.gravy-calculator.de/). The secondary structure of the TaTDRL protein was predicted with the online tool SOPMA (https://npsa-prabi.ibcp.fr/cgi-bin/npsa_automat.pl?page=npsa_sopma.html). Prediction of physical and chemical parameters was done by the ProtParam tool (http://web.expasy.org/protparam/).

### 4.4. RNA Extraction, cDNA Synthesis, and Gene Expression Analysis

Total RNA was extracted from grains, roots, stems, leaves, panicles and treated materials with TRIzol (TaKaRa, Japan). RNA used as templates for cDNA synthesis by TransScript^®^ One-Step gDNA Removal and cDNA Synthesis SuperMix (TransGen Biotech, Beijing, China). qRT-PCR analysis was carried out using the TransStart^®^ Tip Green qPCR SuperMix (TransGen Biotech, Beijing, China) as recommended by the manufacturer. Primers used in the qRT-PCR were 5′-CAACGACCGCCTCTACAAG-3′ and 5′-GCTCCTTCACCTGCTTCTG-3′ for TaTDRL, and 5′-TGTTGTTCTCAGTGGAGGTTCT-3′ and 5′-CTGTATTTCCTTTCAGGTGGTG-3′ for *TaActin*. The *TaActin* was chosen as an internal control. All qRT-PCR amplification was carried out for 30 s at 94 °C, follewed by 42 cycles of 94 °C for 5 s and 60 °C for 30 s. The relative mRNA level for *TaTDRL* was calculated using the 2^−ΔΔt^ method [22]. For qRT-PCR analysis of a chlorophyll-related gene, *AtPORC*, the expression level was detected in the WT and the transgenic plants (TaTDRL-OE and TaTDRL-EAR). The samples for RNA extractions were harvested at the indicated time points (1, 2, 3, and 4 weeks). *AtActin* was used as a reference gene. For each sample, qRT-PCR was performed with three technical replicates from three biological replicate samples.

### 4.5. Promoter Sequence Analysis

The 2000 bp promoter region of the *TaTDRL* was obtained on the EnsemblPlants database. The *TaTDRL* promoter sequence was analyzed using the New PLACE database (https://www.dna.affrc.go.jp/PLACE/?action=newplace) and the PlantCARE database (http://bioinformatics.psb.ugent.be/webtools/plantcare/html/). The analyzed *cis*-elements are listed in Appendix A.

### 4.6. Subcellular Localization and Transcription Activation Activity Assay of TaTDRL

The full-length CDS of TaTDRL was amplified using the gene-specific primers (TaTDRL-EGFP-F: 5′-CGAGCTCAAGCTTCGAAATGGGAGGAGGAGATTATCACC-3′ and TaTDRL-EGFP-R: 5′-CGACTGCAGAATTCGAAATCCATGGCGAGGTACTGCAG-3′). The TaTDRL sequence was cloned into the 35S-EGFP vector with EGFP reporter gene to create a recombinant cassette, wheat 35S::TaTDRL-EGFP. Recombinant plasmids were transiently expressed in *N. benthamiana* leaf cells [38]. After 36 h, tobacco leaf cells were observed on an IX83-FV1200 confocal laser scanning microscope (Olympus). For assaying the transcription activation activity of TaTDRL, the full CDS of TaTDRL cloned by specific primers (5′-GCCATGGAGGCCGAATTCATGGGAGGAGGAGATTATCACC-3′ and 5′-CGGCCGCTGCAGGTCGACTCAATCCATGGCGAGGTACTGC-3′) was constructed into pGBKT7 which was a yeast expression vector. The TaTDRL CDS was divided into five parts and each amplified DNA fragment was inserted into the pGBKT7 (The primers used for developing constructs are listed in Appendix A). These constructs, pGBKT7 (negative control) or pGBKT7-53 (positive control) respectively were transformed into the yeast cell Y2H. Then the transformants were grown on the SD/Trp- and SD/Trp-/His-/Ade-/X-α-gal medium plates. The plates were placed at 30 °C for growth and photographed after three days.

### 4.7. Plant Transformation

The full-length CDS of *TaTDRL* was amplified from XN1376 using TaTDRL-OE-F and TaTDRL-OE-R primers (Appendix A). The PCR product was cloned into the CaMV 35S promoter driven expression cassette of pCAMBIA1302 using NcoI and SpeI restriction sites to generate the 35S::TaTDRL construct. Further, in order to construct the chimeric TaTDRL repressor, we fused the coding region of *TaTDRL* with the DNA encoding for the EAR-motif (LDLDLELRLGFA) repression domain (RD). The sequences were fused in frame, and the resultant polynucleotides were fused with the 35S promoter of CaMV to yield 35S::TaTDRLRD (Appendix A). Both of two constructs were transformed into *Agrobacterium tumefaciens* strain GV3101 for *Arabidopsis thaliana* by the floral dip method [24]. Transgenic lines were grown on 1/2 MS medium supplemented with 50 mg/L hygromycin B. Transcript analysis was performed for the T2 transgenic *Arabidopsis* lines by qRT-PCR. Representative homozygous T3 progeny based on transcript analysis were selected for further studies.

### 4.8. Determination of Chlorophyll Content

The leaves were cut and carefully frozen with liquid nitrogen to determine the chlorophyll content. In the dark room, sample (200 mg) were sliced and incubated in 10 mL of pigment extraction solution (80% acetone) in darkness for 12 h at 25 °C. After centrifugation, the supernatant was collected and absorbance was measured at 663 nm and 647 nm using a spectrophotometer. Contents of chlorophyll a and chlorophyll b in the leaf samples were calculated according to Arnon [39].

### 4.9. Yeast Two-Hybrid Assay

TraesCS5D02G386500.2, namely TaPIF1, was obtained using AtPIF1 (AT2G20180.2) as a query in a BLASTP search of the EnsemblPlants database. Full-length *TaPIF1* and *AtPIF1* CDS were amplified and transformed into the pGADT7 vector. *TaTDRL* CDS without the self-activation fragment ligated into the pGBKT7 vector (The primers are listed in Appendix A). The yeast strain Y2H was then co-transformed with these constructs. Transformants were selected by growing on SD/-Leu/-Trp medium (double drop-out medium, DDO) at 30 °C. Surviving colonies were transferred to DDO and SD/-Ade/-His/-Leu/-Trp medium (quadruple-drop-out medium, QDO) for a further three days before observation.

### 4.10. Bimolecular Fuorescence Complementation (BiFC) Assays

The TaTDRL-nEYFP construct was generated by inserting the N-terminal enhanced YFP (nEYFP) after the full-length *TaTDRL* cDNA without the stop codon in 35S-nEYFP. The TaPIF1-cEYFP construct was made by introducing the C-terminal EYFP (cEYFP) after the full-length TaPIF1 cDNA without the stop codon in 35S-cEYFP. Constructs AtPIF1-cEYFP was created similarly (The primers are listed in Appendix A). All vectors were transformed into GV3101. Agrobacterium cells containing constructs expressing nEYFP fusion proteins and cEYFP fusion proteins were mixed at a ratio of 1:1 and infiltrated into tobacco leaves. BiFC signals were detected by confocal microscopy as described previously [38,40].

### 4.11. Statistical Analysis

Statistical significance was evaluated using Student’s *t*-tests. *p*-values < 0.05 and < 0.01 were used as the thresholds for significant and very significant differences, respectively.

## 5. Conclusions

In conclusion, we identified the *TaTDRL* gene from wheat cultivar XN1376 in this study. Our results demonstrated that TaTDRL is a bHLH protein in wheat and *TaTDRL* expression could be induced by dark treatment. To further reveal the potential functions of TaTDRL, we constructed the *Arabidopsis* overexpression lines of *TaTDRL*. The results demonstrated overexpression of TaTDRL could reduce chlorophyll content, and the results obtained using the CRES-T further confirmed that TaTDRL is a direct inhibitor of chlorophyll synthesis. Our study identified a potential regulator of chlorophyll synthesis and provide a theoretical basis for further reveal of the exact function of TaTDRL.

## Figures and Tables

**Figure 1 ijms-21-00629-f001:**
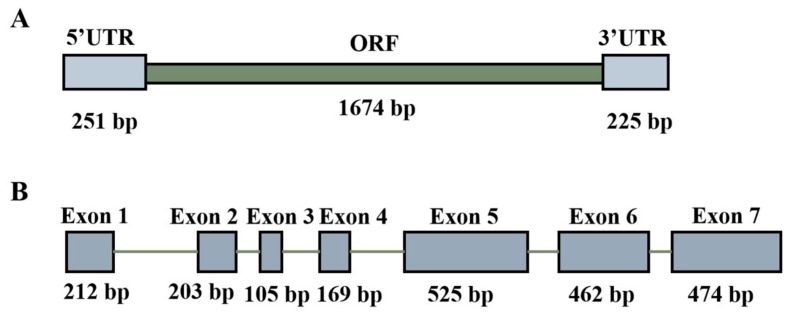
Gene structure analyses for *Triticum aestivum* L. Tapetum Degeneration Retardation-Like (*TaTDRL)*. (**A**) Schematic representation of the corresponding cDNA structure. The 5′ untranslated region (UTR) and 3′ UTR, and the complete coding sequences (CDSs) are indicated by blue and green boxes, respectively. (**B**) Genomic organization of *TaTDRL*. Blue boxes indicate exons encoding amino acids and green lines indicate introns. Numbers represent the lengths of each region in base pairs.

**Figure 2 ijms-21-00629-f002:**
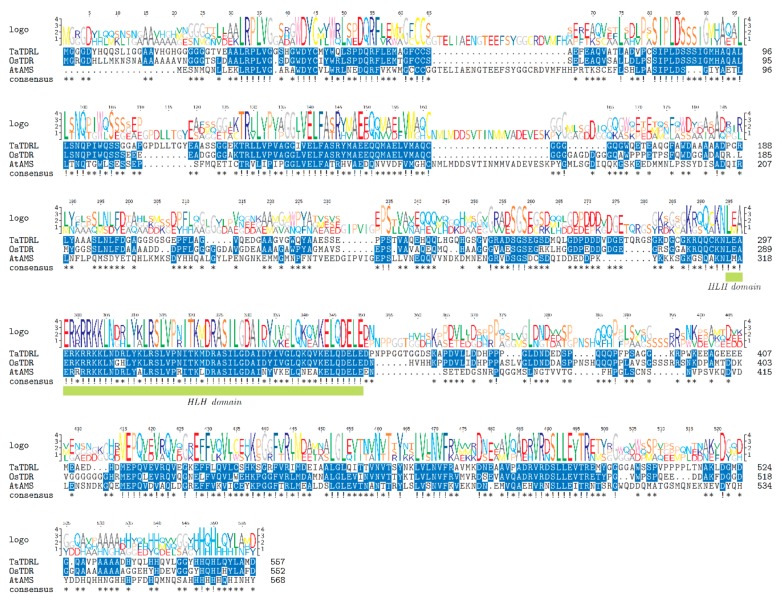
Protein sequence analysis of TaTDRL protein. Multiple alignment of TaTDRL and other basic helix-loop-helix (bHLH) domain proteins comprising *Arabidopsis thaliana* aborted microspores (AtAMS) and rice *Oryza sativa* L., Tapetum Degeneration Retardation (OsTDR). The highly conserved amino acid residues among the proteins examined are shaded. The solid green line shows the bHLH domain (295–344 aa).

**Figure 3 ijms-21-00629-f003:**
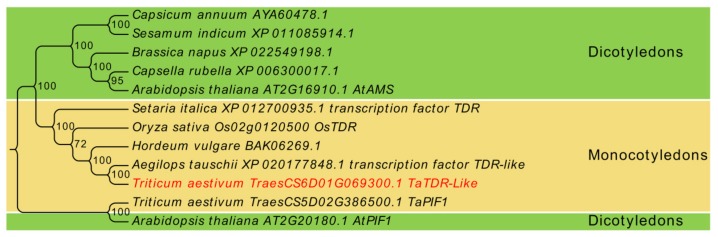
Phylogenetic analysis of TaTDRL. Phylogenetic tree analysis of TaTDRL using Neighbor-joining method using MEGA-X program. Bootstrap values from 1000 replicates were indicated at each node.

**Figure 4 ijms-21-00629-f004:**
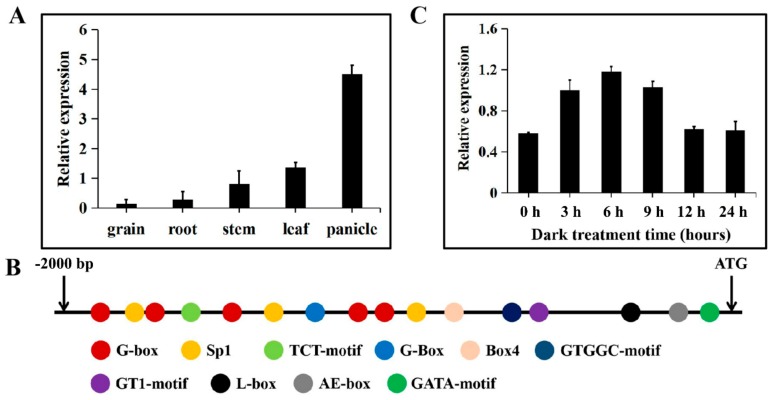
Expression pattern and promoter cis-regulatory element analysis of *TaTDRL*. (**A**) Tissue-specific expression of *TaTDRL* in grain, root, stem, leaf, and panicle tissues in wheat. (**B**) Putative 16 light responsive elements in the promoter region of *TaTDRL* were predicted by PlantCARE. (**C**) Expression patterns of *TaTDRL* in wheat leaves under dark stress for 3, 6, 9, 12, and 24 h, respectively. *TaActin* was used as an internal control. Data represent mean ± SD (*n* = 3).

**Figure 5 ijms-21-00629-f005:**
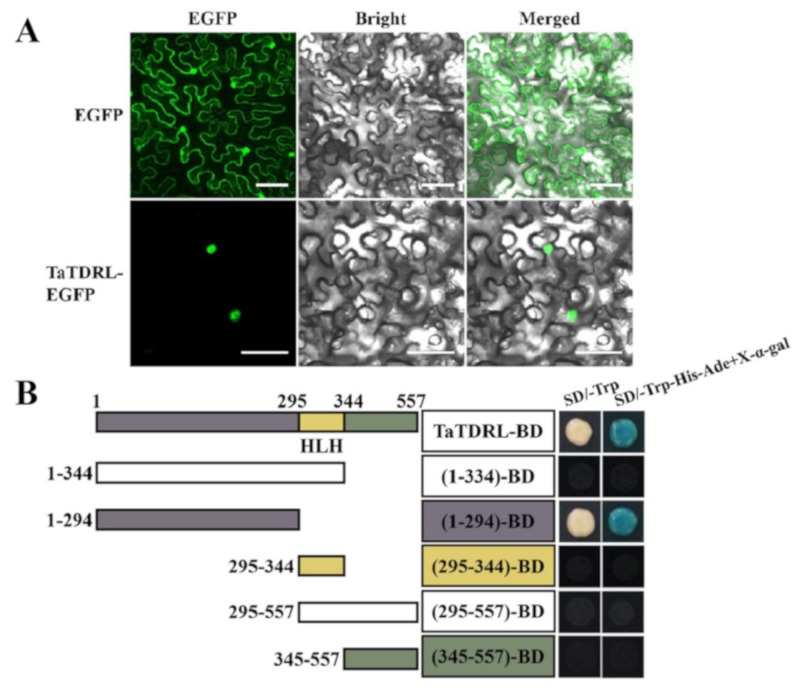
Nuclear-localized TaTDRL has transcriptional activation activity in yeast cells. (**A**) Subcellular localization of TaTDRL-enhance green fluorescence protein (EGFP) fusion protein in tobacco epidermal cells. Scale bars = 50 μm. (**B**) Transactivation assay of the TaTDRL proteins. Full-length and different portions of TaTDRL were fused with the GAL4 DNA-binding domain and then expressed in yeast strain Y2H. The transformed yeast cells were plated and grown on control plates (SD/-Trp) or selective plates (SD/-Trp-His-Ade + X-α-gal).

**Figure 6 ijms-21-00629-f006:**
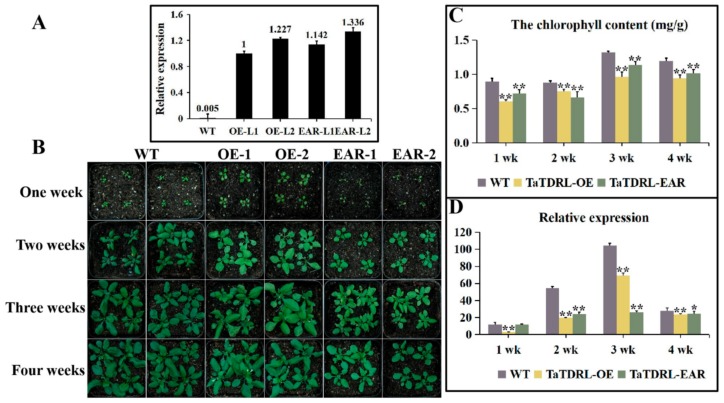
Phenotypic characterization of transgenic and wild-type (WT) *Arabidopsis* plants. (**A**) qRT-PCR identification of transgenic *TaTDRL* overexpression *Arabidopsis* plants. (**B**) Representative images show WT and transgenic lines, after one week of growth, two weeks of growth, three weeks of growth, and four weeks of growth. (**C**) Total content of chlorophyll in WT, TaTDRL-overexpressing (OE) and TaTDRL-EAR under normal conditions. (**D**) Expression of chlorophyll-responsive gene, *AtPORC*, in WT, TaTDRL-OE, and TaTDRL-EAR in response to chlorophyll biosynthesis. *AtActin* was used as an internal control. Data represent means ± SD (*n* = 3). Each column represents the mean ± standard error based on three biological repeats. Significant differences between WT and transgenic *Arabidopsis* lines were determined by Student’s test (* *p* ≤ 0.05, ** *p* ≤ 0.01).

**Figure 7 ijms-21-00629-f007:**
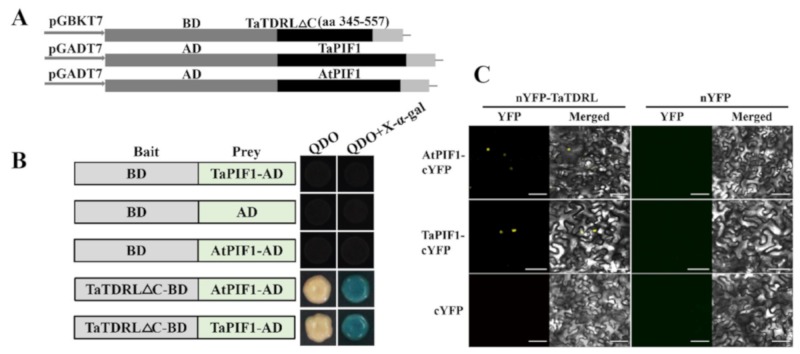
Y2H and Bimolecular Fuorescence Complementation (BiFC) assays to test the interactions of *Arabidopsis thaliana* Phytochrome Interacting Factor 1 (AtPIF1) and TaPIF1 with TaTDRL. (**A**) The full-length AtPIF1 and TaPIF1 CDS were fused with the GAL4 activation domain (AD) pGADT7, and TaTDRL CDS without the self-activation fragment (TaTDRL∆C) was fused with the DNA-binding domain (BD) in pGBKT7 as the construction schematic diagrams showed. (**B**) Yeast two-hybrid assay showing the interaction of AtPIF1 and TaPIF1 with TaTDRL. Yeast cells coexpressing the indicated combinations of constructs were grown on selective QDO (SD\-Trp\-Leu\-His\-Ade). The interactions (represented by blue color) were assessed on QDO + X-α-Gal medium. (**C**) BiFC assays showing the interaction of AtPIF1 and TaPIF1 with TaTDRL in tobacco. Scale bars = 80 μm.

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
