# Peer review of "Isolation and Identification of a TaTDR-Like Wheat Gene Encoding a bHLH Domain Protein, Which Negatively Regulates Chlorophyll Biosynthesis in Arabidopsis"

_ijms, 2020, doi:10.3390/ijms21020629_

Round 1

Reviewer 1 Report

Dear Editor

In their manuscript, Dr. Yu Xia and collaborators describe the functional characterization of a Triticum aestivum gene in Arabidopsis thaliana plants. The gene encodes a putative bHLH domain protein, named TaTDR-like, orthologue of the Oriza sativa Tapetum Degeneration Retardation (TDR) gene, previously proved to be implicated in tapetum degradation and anther development (Na Li et al., 2006).

Here the authors proved that TaTDR-Like gene encodes a protein targeted to the nucleus and acts a transcription factor. By overexpressing the gene in Arabidopsis, they found an effect on chlorophyll content. By using a Chimeric REpressor gene Silencing Technology, they produced Arabidopsis lines, expressing the chimeric TaTDR-EAR repressor transcription factor, characterized by a lower chlorophyll content as compared with WT plants. By direct Y2H and BIFC, TaTDRL gene product was proved to interact with both Arabidopsis and wheat phytochrome-interacting factor (PIF) protein.

The experiments are generally well conducted with the proper controls and the material and methods section detailed.

I have some criticisms, and I would like to have some clarifications on these points. Here a list of the minor and major criticisms:

I think that the term “transgenic” could be removed from the title, because it seems obvious to me. Abstract: line 17, please specify how you identified this gene (e.g. transcriptomic analysis, sequence comparison…etc); line 22 (it is not correct “we identified the role of TaTDRL in wheat”). On this specific point, I think that the authors should better emphasise in the text that the functional study conducted in transgenic Arabidopsis revealed the potential function of TaTDRL gene in wheat. However, we cannot ignore the differences between Arabidopsis and wheat, consequently a detailed functional analysis of TATDRL gene using transgenic wheat will be necessary to prove its complete function. This point needs to be better highlighted in the text (I think in the discussion and/or by introducing a brief sentence at the end of the abstract). Line 24: please specify with the acronyms of the lines the term “both of two T3 transgenic lines”. Introduction: text between lines 35-44 could be shortened. I think that in the introduction a more detailed description on the role of OsTDR gene should be reported (here restricted only to line 78). In addition, lines 222-241 could be more informative for the readers if moved to the introduction section. Results:Paragraph 2.1: Figure 3; please specify in the legend the name of the genes and the gene ID where not clearly reported (I’m referring in particular to the Arabidopsis thaliana genes). In addition, it would be better to define with two different colour the name of the species from monocots and dicots. In my opinion it is not clearly described the rationale behind the analysis of TaTDRL gene in chlorophyll biosynthesis, considering the previously published study on reproductive development (i.e. line 78-79 of the introduction section) also considering that the highest expressions in wheat, as reported by the authors in Figure 4a, is detected in panicle (inflorescence). Paragraph: 2.4: Due to the high sequence similarity between OsTDR and TsTDRL, and considering the highest expression of the gene in the inflorescences, I would expect phenotypic alterations in Arabidopsis plants for what concerns the reproductive development (What’s about the anther development? Have the authors observed alterations in seed number?). I think that this is a crucial point and needs to be better detailed. Paragraph 2.2: why only a dark stress has been applied to test the light responsiveness? Paragraph 2.6 Please introduce a brief explanation to describe the reasons for AtPORC choice. The introduction is too distant for a reader not familiar with this gene to remember it. I’ve noticed that the authors accidentally use several acronyms somewhere before defining their entire name. Please check in manuscript that each acronym is properly explained.

Reviewer 2 Report

The manuscript evaluated shows an interesting multi approach to prove the identity of TaTDR-like in wheat as a negative regulator of chlorophyl biosynthesis.

It´s clear and interesting, and the manuscript flows in a proper manner from the identity and phylogenetic context of the protein, to the function and localization at subcellular level.

Major concerns, out of minor comments included in the revised attached file, are:

> it is claim that the protein is light induced, but no experimental data supported the assumption, I would recommend a simple assay colelcting samples after light treatments and evaluate it´s expression levels by qPCR.

> heterologous complementation is only performed by overexpressor constructions, why not to use arabidopsis native promoter with wheat CDS?

> colour of leaves is discussed, but not quantitative data presented (i.e. ImageJ analysis)

> proved of overexpressor lines should be confirmed by quantitative qPCR

Minor comments are included on the attached file.

Round 2

Reviewer 1 Report

Dear Editor

the manuscript has been improved and in my opinion it meets the quality criteria for publication.

Reviewer 2 Report

I am satisfied with the modifications and responses from the authors, so no main concerns for publication on its current form.